# A Lightweight Military Target Detection Algorithm Based on Improved YOLOv5

**Xiuli Du** [1,2,*], **Linkai Song** [1,2], **Yana Lv** [1,2] **and Shaoming Qiu** [1,2]

1   Communication and Network Laboratory, Dalian University, Dalian 116622, China
2   School of Information Engineering, Dalian University, Dalian 116622, China
*   Correspondence: duxiuli@dlu.edu.cn

**Abstract:** Military target detection technology is the basis and key for reconnaissance and command decision-making, as well as the premise of target tracking. Current military target detection algorithms involve many parameters and calculations, prohibiting deployment on the weapon equipment platform with limited hardware resources. Given the above problems, this paper proposes a lightweight military target detection method entitled SMCA-$\alpha$-YOLOv5. Specifically, first, the Focus module is replaced with the Stem block to improve the feature expression ability of the shallow network. Next, we redesign the backbone network of YOLOv5 by embedding the coordinate attention module based on the MobileNetV3 block, reducing the network parameter cardinality and computations, thus improving the model's average detection accuracy. Finally, we propose a power parameter loss that combines the optimizations of the EIOU loss and Focal loss, improving further the detection accuracy and convergence speed. According to the experimental findings, when applied to the self-created military target data set, the developed method achieves an average precision of 98.4% and a detection speed of 47.6 Frames Per Second (FPS). Compared with the SSD, Faster-RCNN, YOLOv3, YOLOv4, and YOLOv5 algorithms, the mAP values of the improved algorithm surpass the competitor methods by 8.3%, 9.9%, 2.1%, 1.6%, and 1.9%, respectively. Compared with the YOLOv5 algorithm, the parameter cardinality and computational burden are decreased by 85.7% and 95.6%, respectively, meeting mobile devices' military target detection requirements.

**Keywords:** military target detection; YOLOv5; Stem block; MobileNetV3 block; coordinate attention; loss function

## 1. Introduction

Military target detection technology is the key to improving battlefield situation generation, reconnaissance, surveillance, and command decision-making and is an essential factor for winning modern warfare. Real-time and accurate detection of battlefield targets will help us grasp the battlefield environment faster, search and track enemy units, and understand the enemy's dynamics to seize the opportunity in the war and be in a dominant position [1–3].

A large amount of data, rapid changes, and strong camouflage are features of battlefield targets in modern combat, influenced by artificial intelligence's development [4,5]. Most traditional visual target detection technologies are based on hand-designed features for target detection, and it is challenging to obtain target information comprehensively, quickly, and accurately from the complex battlefield environment.

Computer vision technology has become widely used in various industries, including video surveillance, drone piloting, and military intelligence analysis, due to the rapid growth of deep learning [2]. Currently, target detection algorithms based on deep learning can be divided into candidate frame-based and regression-based algorithms. The former is represented by the Region-based Convolutional Neural Network (R-CNN) [6], Fast R-CNN [7], and Faster R-CNN [8]. The latter mainly include You Only Look Once (YOLO)

series algorithms [9–12] and SSD [13–15] algorithms. In order to achieve a higher detection accuracy, the target detection algorithm based on the candidate frame first counts the target frame on the feature map and then obtains the detection result in a refined manner. However, there are drawbacks, such as high memory resource consumption and slow speed. The regression-based target detection algorithm is an end-to-end detection method. The target is obtained by direct regression on the feature map, so the detection speed is significantly improved, but the detection accuracy is slightly lower than that of the candidate frame-based detection algorithm.

Several scholars have successfully applied deep learning-based methods in military target detection recently. For instance, [16] proposed a neural network-based military vehicle detection method, attaining a recognition rate of 97.36%. In [17], the authors proposed an improved Fast R-CNN algorithm for small tank target detection. This algorithm is superior to the Faster R-CNN algorithm in detection speed and accuracy but suffers from miss-detections when detecting occluded targets. The work of [18] suggested a tank military robot with target detection and tracking functions, effectively improving the battlefield's combat capability. Reference [19] proposed a remote sensing image selection and searching method to solve the potential hot spot detection problem in large-scale remote sensing images and improve the detection accuracy of overlapping targets. This method improves the target detection accuracy without considering the model's space complexity. In [20], the authors fully integrated polarization imaging and deep learning to detect camouflaged artificial targets quickly under normal and low illumination conditions. Reference [21] developed a new military target detection algorithm, which introduced the GhostNet module to improve the detection accuracy and speed and then improved the loss function to enhance detection accuracy. The experimental results show that the model's parameters are about three times higher than the YOLOv5 model. Furthermore, reference [22] solved the DIOU defect when the center of the bounding box was aligned at the same point, which is conducive to the efficient deployment of detection algorithms in resource-constrained environments. Reference [23] proposed an armored target detection algorithm named GCD-YOLOv5 that utilized a LIDAR array in complex environments. This algorithm has a strong detection ability, but its network structure is complex and thus challenging to implement the transplantation of embedded terminals.

Based on the above research, with the continuous improvement in the performance of the network model, the increase in the number of model parameters and computation restricts its embedding in resource-constrained weapons and equipment. In order to meet the requirements of military target detection under limited resources of weapon hardware platforms, this paper proposes an improved YOLOv5 algorithm (SMCA-$\alpha$-YOLOv5), which is tested and compared through ablation experiments. The results show that compared with YOLOv5s, the mean average precision is increased by 1.9%, the amount of model parameters is decreased by 85.7%, and the amount of computation is decreased by 95.9%. The main contributions of this paper can be summarized as follows:

1.  The Stem block is used to replace the Focus module, and the multi-channel information fusion improves the feature expression ability, reducing the model's parameters and computation complexity.
2.  The coordinate attention module is embedded in the MobileNetV3 block structure to redesign the backbone network of YOLOv5. This strategy reduces the network's parameters and computation complexity and improves its detection performance.
3.  Considering the defects of CIOU loss, we propose a power parameter loss optimized by combining the EIOU loss and Focal loss. The experiments show that the convergence speed is faster and the regression error is lower.

The remainder of this paper is organized as follows: Section 2 introduces the construction of the military target dataset. Section 3 introduces the work related to the YOLOv5s structure, MobileNetV3 block module, coordinate attention mechanism, and Loss Metrics in Object Detection. Section 4 introduces the improved YOLOv5 algorithm.

Section 5 analyzes and discusses the experimental results. Finally, Section 6 presents the conclusion and future work.

## 2. Datasets

With the vigorous development of deep learning, the performance of target detection algorithms based on deep learning depends on the quality of large-scale data sets. Therefore, preparing large-scale military target data sets is the basis and premise of research on military target detection. The current mainstream target detection datasets mainly include PASCAL VOC [24], MS COCO [25], ImageNet, etc. These datasets mainly include common objects such as furniture, electronic equipment, vehicles, and people, and some datasets contain tanks, soldiers, and military targets such as drones, but the data types are small, the amount of data is insufficient, and the background is simple. Due to the particularity of the types of military targets, and confidentiality considerations, the public dataset resources are relatively small, and it is difficult to train deep neural networks. Therefore, this paper makes the Military Image Target Dataset (MITD).

### 2.1. Source of Data

Military objectives can be divided into sea, land, and air. Maritime military targets mainly refer to submarines, naval warships, etc.; land military targets mainly include tanks, soldiers, trucks, and other weapons and equipment; air military targets mainly include helicopters, early warning aircraft, missiles, etc. [3]. This article obtained 9369 military target images in jpg through the Google search engine. It mainly includes seven military targets: tank, missile, helicopter, air early warning, ship, submarine, and soldier. In this paper, all kinds of targets in the military target dataset are randomly divided into a training set, validation set, and test set according to 7:2:1. Figure 1 shows a sample example of MITD.

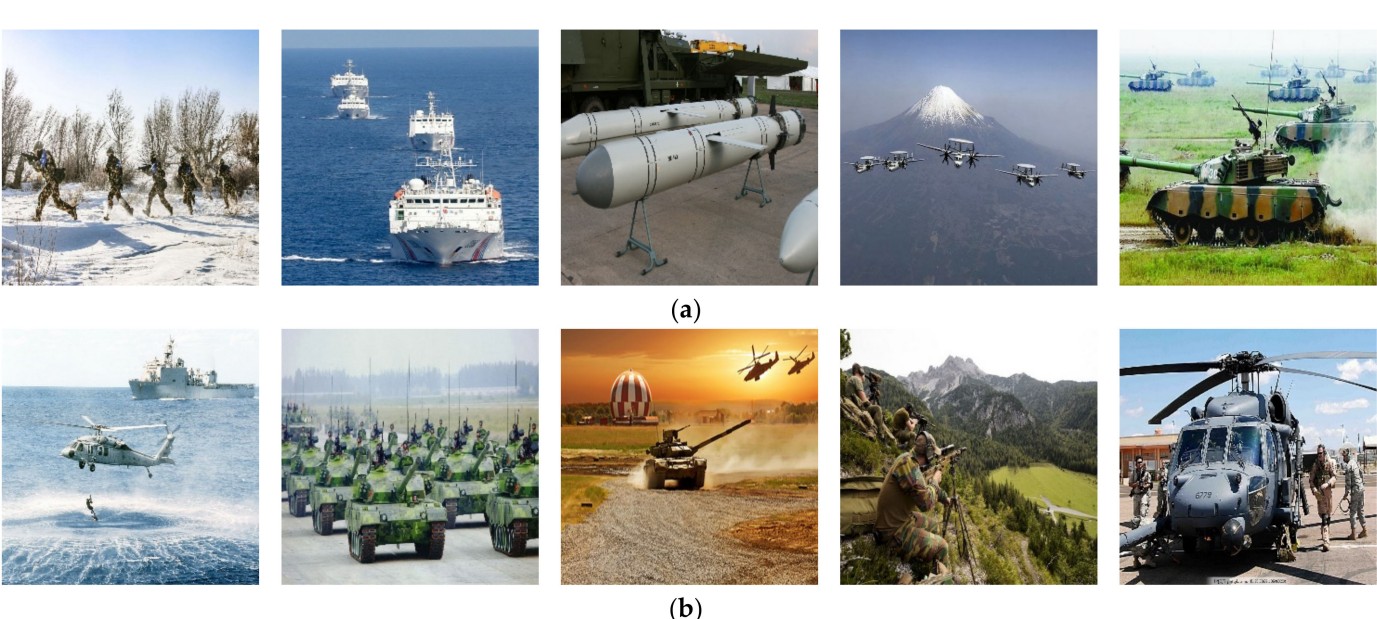

**Figure 1.** Sample images in MITD. (**a**) A picture contains a single military target; (**b**) A picture contains multiple military targets.

### 2.2. Label Format and Data Size

In this paper, labeling software is used to label the image targets in MITD, and the target's position information is stored in the text document in YOLO format, and 9369 text documents are finally obtained. Statistical analysis of various military target information is shown in Table 1. There are 9369 images, including 13,199 target boxes, and the number of

targets ranges from 1 to 16. The image width pixel range is [233, 4960], and the height pixel range is [167, 2802].

**Table 1.** Details of the MITD.

| Military Target Name | Image | | | Target Box |
|---|---|---|---|---|
| | Number | Range of Heights | Range of Width | Number |
| tank | 1448 | [233, 2560] | [180, 1600] | 1667 |
| missile | 1441 | [273, 4096] | [180, 2800] | 1911 |
| helicopter | 986 | [267, 3000] | [230, 2000] | 1126 |
| air early warning | 1456 | [400, 3100] | [260, 2063] | 1591 |
| submarine | 1717 | [399, 2500] | [262, 2437] | 1754 |
| warship | 1394 | [240, 4960] | [240, 2802] | 1901 |
| soldier | 927 | [266, 2048] | [167, 1360] | 3249 |
| total/range | 9369 | [233, 4960] | [167, 2802] | 13,199 |

## 3. Related Work

This section will introduce the related principles of YOLOv5, MobileNetV3 block, coordinate attention mechanism, and Loss Metrics in Object Detection.

### 3.1. YOLOv5 Algorithm

The YOLOv5 algorithm [26–29] is an open-source object detection project with good engineering results. At present, four versions of YOLOv5s, YOLOv5m, YOLOv5l, and YOLOv5x are included in the released YOLOv5 project. Among them, the YOLOv5s structure is the network with the smallest depth and width, and has the advantages of high speed and small size. Therefore, this paper adopts the YOLOv5s structure, which consists of four parts: the Input, the Backbone network, the Neck network layer, and the Head output, as shown in Figure 2.

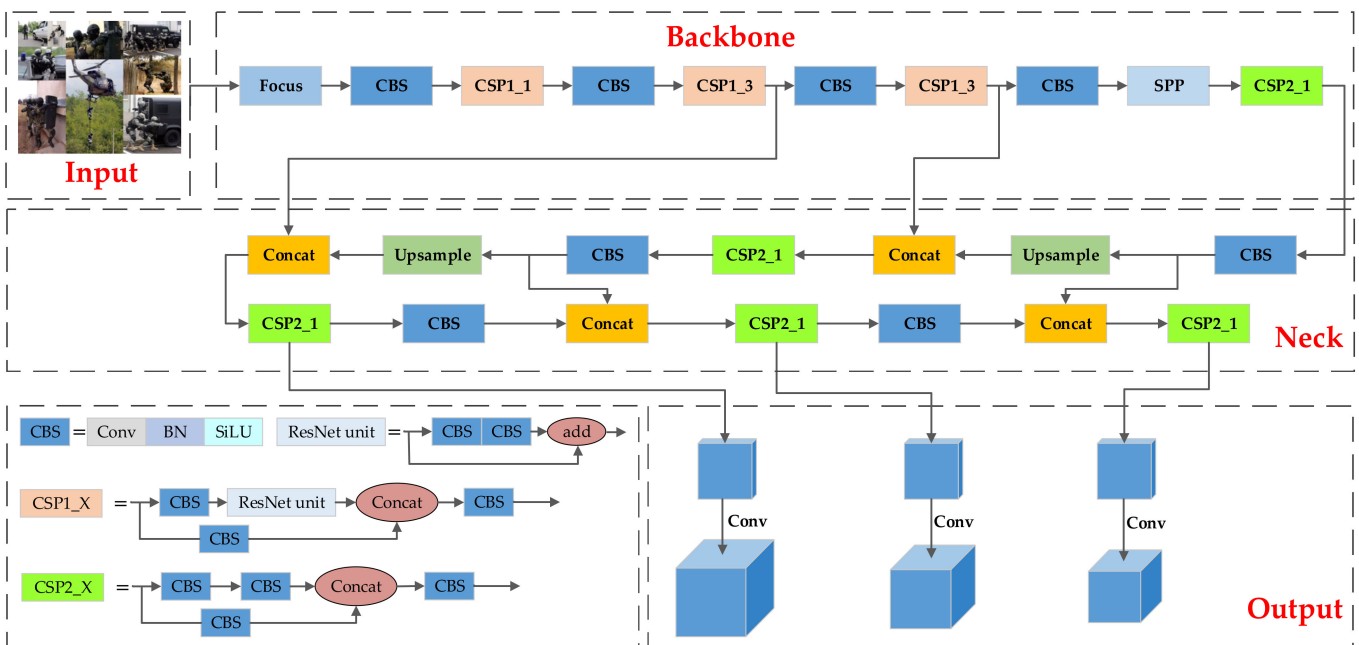

**Figure 2.** The structure of YOLOv5s.

- Input: The Input preprocesses the original image data, mainly including Mosaic data enhancement, random cropping, and adaptive image filling. In order to adapt to different target data sets, adaptive frame calculation is integrated into the input.

- Backbone: The Backbone network extracts the feature information at different levels of the image through the deep residual structure. The main structures are Cross Stage Partial (CSP) [30] and Spatial Pyramid Pooling (SPP) [31]. The former aims to reduce the amount of calculation and improve the inference speed. The latter aims to perform feature extraction at different scales for the same feature map, which helps to improve detection performance.
- Neck: The Neck network layer includes Feature Pyramid Networks (FPN) and Path Aggregation Network (PAN). FPN transmits semantic information from top to bottom in the network, and PAN transmits positioning information from top to bottom. The information is fused to improve the detection performance further.
- Head: The head output uses the feature information extracted from the Neck end to filter the best detection frame through non-maximum suppression, and generates a detection frame to predict the target category.

### 3.2. MobileNetV3 Block

The MobileNet algorithm series is representative of lightweight network models. MobileNetV1 [32] introduced depthwise separable convolution instead of standard convolution and reduced the number of parameters and computation of the model through the combination of channel-by-channel convolution and point-by-point convolution. MobileNetV2 [33] draws on the residual network to design an inverse residual structure that first increases the dimension, performs convolution, and then reduces the dimension. At the same time, it uses a linear bottleneck layer structure to retain the effective features to the greatest extent, and it is easy for the model to be deployed on mobile devices. MobileNetV3 [34] introduces a lightweight attention mechanism SENet based on MobileNetV2, uses the improved swish activation function to upgrade the nonlinear layer, and finally uses the neural network architecture to search for the best network model. The unit module is shown in Figure 3.

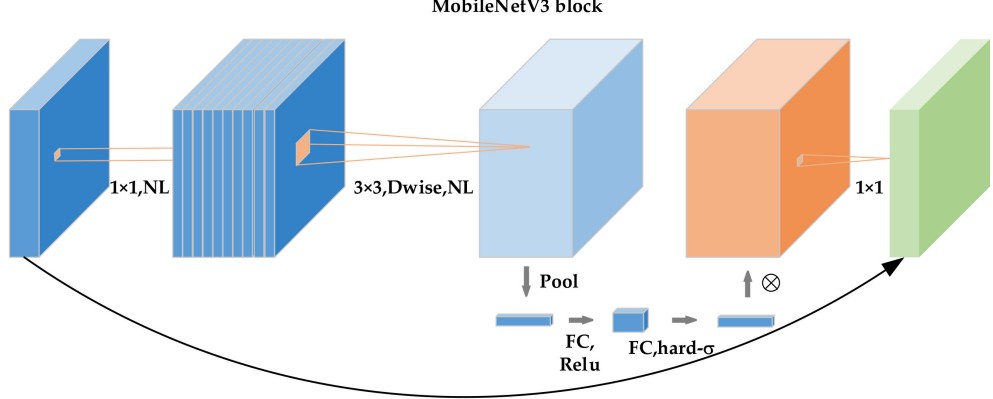

**Figure 3.** The structure of MobileNetV3 block. NL denotes the type of nonlinearity used.

### 3.3. Coordinate Attention

The coordinate attention (CA) mechanism [35] embeds location information into channel attention, decomposing the channel attention into a one-dimensional encoding process that aggregates feature along two spatial directions. Long-term dependencies can be captured in one spatial direction, while precise location information can be preserved in the other. The representation of objects of interest can be improved by applying a pair of orientation-aware and position-sensitive feature maps in addition to the input feature maps. The fundamental purpose of a coordinate attention block is to enhance the expressive ability of mobile network learning features. As shown in Figure 4, this module is divided into embedding collaborative information and generating coordinated attention. First, it arbitrarily takes two intermediate feature tensors $X = [x_1, x_2 \cdots , x_C] \in R^{C \times H \times W}$ and $Y = [y_1, y_1, \cdots , y_1] \in R^{C \times H \times W}$, where $X$ is the input and $Y$ is the output. The embedding

of collaborative information is given input $X$, using multiple pooling kernels for $(H,1)$ and $(1,W)$ to encode channels along the direction and longitudinal direction, respectively. Therefore, the output of the height $h$ and $c^{th}$ channel can be expressed as:

$$Z_c^h(h) = \frac{1}{W} \sum_{0 \leq i \leq W} x_c(h,i) \tag{1}$$

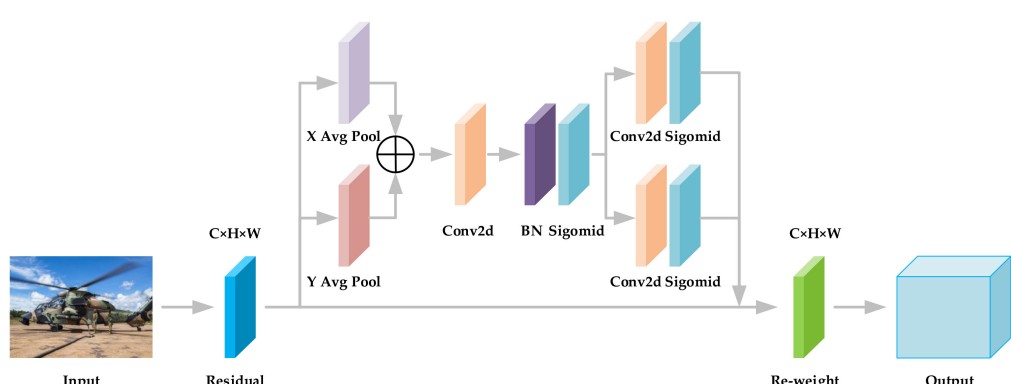

**Figure 4.** The structure of the Coordinate Attention block.

Similarly, the output of the width $w$ and $c^{th}$ channel can be expressed as:

$$Z_c^w(w) = \frac{1}{H} \sum_{0 \leq j \leq H} x_c(j,w) \tag{2}$$

The two transformations, Equations (1) and (2), aggregate features along two spatial directions, respectively. The generation of collaborative attention is the splicing of the two transformations, and then sent to a transformation $F_1$ that shares the $1 \times 1$ convolution, which can be expressed as:

$$f = \delta\left(F_1\left(\left[z^h, z^w\right]\right)\right) \sum_{0 \leq i \leq W} x_c(h,i) \tag{3}$$

In Equation (3), $\left[z^h, z^w\right]$ is the splicing operation along the spatial dimension, $\delta$ is the nonlinear activation function, and $f \in R^{C/r \times (H+W)}$ is the horizontal Intermediate feature maps that encode spatial information in the directional and vertical directions. $r$ is the reduction ratio. $f$ is split into two separate tensors $f^h \in R^{C/r \times H}$ and $f^w \in R^{C/r \times W}$ along the spatial dimension. In addition, using two $1 \times 1$ convolutional transforms $F_h$ and $F_w$ to transform $f_h$ and $f_w$ into tensors with the same number of channels for the input $X$ yields:

$$g^h = \delta\left(F_h\left(f^h\right)\right) \tag{4}$$

$$g^w = \delta(F_w(f^w)) \tag{5}$$

$\delta$ is a sigmoid function. Extend the outputs $g^h$ and $g^w$. Finally, the output of $Y$ can be written as:

$$y_c(i,j) = x_c(i,j) \times g_c^h(i) \times g_c^w(j) \tag{6}$$

### 3.4. Loss Metrics in Object detection

Bounding box regression (BBR) is a critical step in object detection techniques, and a well-designed loss function is crucial to the success of BBR. Currently, most detection methods use BBR, and the loss function of BBR can be roughly divided into horizontal and rotational detection regression loss.

Researchers have carried out much work on designing level detection regression loss functions. For instance, YOLOv1 [9] proposes the square root of the predicted bounding box size to compensate for the difference between scales. Fast R-CNN [7] and Faster R-CNN [8] use the $l_1$ loss function, which is less sensitive to outliers than the $l_2$ loss. However, most of the $l_n$ norm loss functions assume that the bounding box variables are independent, which is inconsistent with the real situation. In response to the above problems, the IOU [36] loss was proposed, achieving good performance then. To address the shortcoming of IOU loss, the Generalized IOU (GIOU) [37] loss was proposed, i.e., the IOU loss is always zero when the two boxes do not overlap. In order to further solve the shortcomings of GIOU's slow convergence speed in the horizontal and vertical directions, Distance IOU (DIOU) and Complete IOU (CIOU) [38] have been proposed, with experiments demonstrating that these two losses converge faster and have better performance. Since the aspect ratio of the CIOU loss is relative and does not balance the hard and easy samples, the EIOU loss and the Focus-EIOU loss [39] are proposed. In order to achieve a more flexible accuracy of the bounding box regression at different levels, an Alpha-IoU loss was proposed [40].

The above IOU loss only applies to the simple axis alignment case and cannot be directly applied to rotation detection. Thus, [41] studied the IoU computation of two rotating boxes and implemented a unified framework for 2D and 3D object detection tasks. The PIoU method [42] does this by simply counting the number of pixels. Furthermore, to address the convex uncertainty caused by rotation, [43] proposed a projection operation to estimate the intersection area and [44] developed a new regression loss based on the Gauss Wasserstein distance to solve the boundary discontinuity and detection index inconsistency problems in the design of rotation detection regression loss.

## 4. Approach

This section details the improvement methods of YOLOv5, including the introduction of the Stem block, the design of the MNtV3-CA module, the optimization of the loss function, and the overall structure design of the network.

### 4.1. Introduction of Stem Block

Military target detection not only puts forward higher requirements on the detection accuracy and detection speed of the target but also is affected by the limitations of the memory and computing resources of the weapon equipment platform. The Focus module of the YOLOv5 algorithm improves the detection speed of the model to a certain extent but greatly increases the amount of calculation and parameters.

Therefore, designing a military target detection algorithm with small memory and less computation is very important. With the above requirements, this paper introduces the Stem block structure, as shown in Figure 5. This structure has achieved good results in real-time detection algorithms on mobile devices, such as PELEE [45], PP-LCNet [46], YOLO5Face [47], etc. Inceptionv4 and Deeply Supervised Object Detector inspire the design of the Stem block. By replacing the large convolution module with a smaller computation cost and parameters, the module improves feature expression ability with almost no increase in computation and parameters.

### 4.2. MNtV3-CA Block Structure

The backbone network of the YOLOv5 algorithm adopts the traditional residual structure, which solves the problem of network degradation caused by the increase in the network structure's depth, and has a faster convergence speed under the same number of network layers [48]. Residual networks have been widely used in deep neural networks, improving the network performance by increasing the network depth. However, this substantially increases the network parameters, making it difficult to train the model. It is not easy for the network to calculate Deploy on weapons with limited capabilities and memory resources. Therefore, this paper designs a lightweight MNtV3-CA structure to redesign the backbone network of the YOLOv5 algorithm, as shown in Figure 6. This

structure is based on the MobliNetV3 block and integrates the lightweight coordinate attention module, enhancing the model's detection performance while ensuring a light network structure.

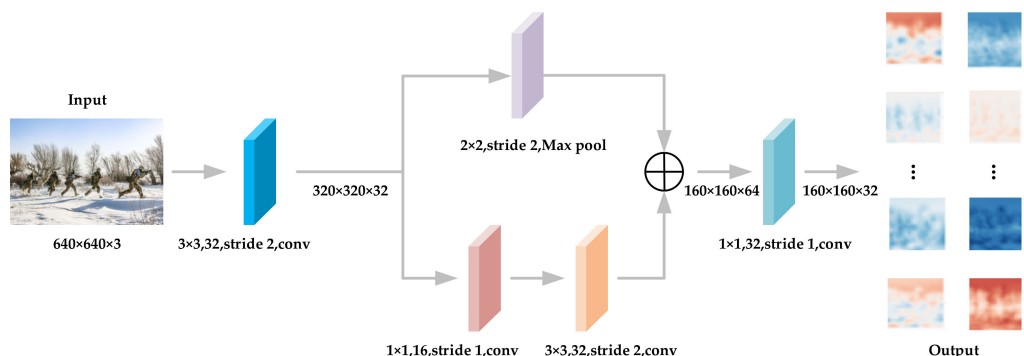

**Figure 5.** The structure of Stem block.

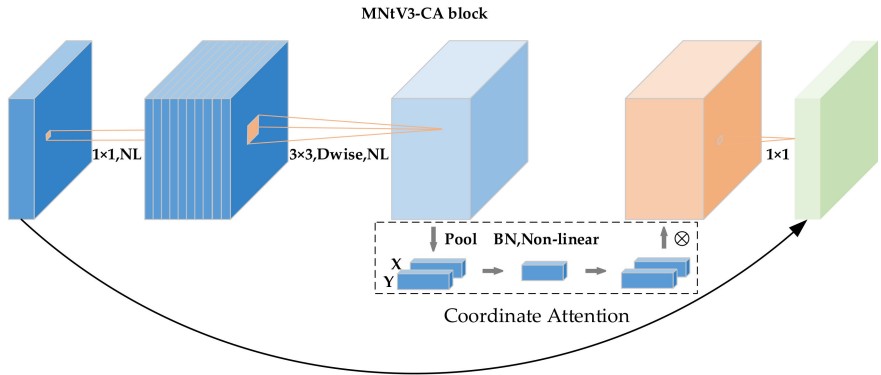

**Figure 6.** The structure of MNtV3-CA block.

### 4.3. Optimization of Loss Function

The IOU function is the most commonly used evaluation index in the field of target detection, used to measure the overlap rate between the target box and the predicted box, $A, B \subseteq S \in R^n$, where $A$ represents the area of the target box, and $B$ represents is the predicted box area. The formula is as follows:

$$IOU = \frac{|A \cap B|}{|A \cup B|} \tag{7}$$

The YOLOv5 algorithm uses the CIOU loss [38], which considers three important geometric factors: the overlap between the predicted box and the target box, the distance between the center points, and the aspect ratio. The disadvantage is that $v$ in the formula only reflects the difference in aspect ratio, which increases the similarity of aspect ratio to a certain extent, but sometimes hinders the real difference between aspect ratio and confidence and does not consider the balance of difficult and easy samples [39].

To solve the shortcomings of the CIOU loss, this paper introduces the EIOU loss [39], which improves the CIOU loss by discarding the penalty term of the aspect ratio and employing the prediction results of width and height to guide the loss convergence. EIOU loss is formulated as:

$$
\begin{aligned}
L_{EIOU} &= L_{IOU} + L_{dis} + L_{asp} \\
&= 1 - IOU + \frac{\rho^2(b, b^{gt})}{c^2} + \frac{\rho^2(w, w^{gt})}{C_w{}^2} + \frac{\rho^2(h, h^{gt})}{C_h{}^2}
\end{aligned} \tag{8}
$$

where $C_w^2$ and $C_h^2$ are the width and height of the minimum circumscribed rectangle of the prediction box and the target box, respectively, $\rho^2(\cdot)$ is the Euclidean distance between the prediction box and the target box, $b$ is the center point of the prediction box, $b^{gt}$ is the center point of the target box, $w$ and $h$ is the width and height of the prediction box, respectively, and $w^{gt}$ and $h^{gt}$ are the width and height of the target box, respectively.

Equation (8) reveals that the EIOU loss is divided into three parts: the IOU loss $L_{IOU}$, the distance loss $L_{dis}$, and the aspect loss $L_{asp}$. The EIOU loss not only retains the characteristics of the CIOU loss but also reduces the difference between the width and height of the target box and the anchor box, affording a rapid model convergence and accuracy improvement. Inspired by Alpha-IoU [40], this paper generalizes EIOU loss to a loss function with power terms, defined as α-EIOU loss, formulated as:

$$L_{\alpha-EIOU} = 1 - IOU^{\alpha} + \frac{\rho^{2\alpha}\left(b, b^{gt}\right)}{c^{2\alpha}} + \frac{\rho^{2\alpha}\left(w, w^{gt}\right)}{C_w{}^{2\alpha}} + \frac{\rho^{2\alpha}\left(h, h^{gt}\right)}{C_h{}^{2\alpha}} \tag{9}$$

where α is the power parameters.

The Focus-EIOU loss cannot flexibly achieve the accuracy of different levels of the bounding box regression, and the Alpha-IoU does not consider the problem of difficult and easy sample balance. Therefore, this paper combines Focus Loss with the α-EIOU by using the IOUα to weight α-EIOU. This scheme is defined as Focal-α-EIOU Loss and is formulated as:

$$L_{Focal-\alpha-EIOU} = IOU^{\alpha\gamma} L_{\alpha-EIOU} \tag{10}$$

When α = 1, Equation (10) is $L_{Focal-EIOU} = IOU^{\gamma} L_{EIOU}$, and $\gamma$ is a parameter that controls the degree of outlier suppression.

In summary, the proposed Focal-EIOU loss with the power alpha function has the following advantages: (1) adjusting α provides the detector more flexibility to achieve different levels of box regression accuracy, (2) considers the difficulty Easy sample balance problem, and (3) the regression loss is lower, and the convergence speed is faster.

### 4.4. Network Structure of SMCA-α-YOLOv5

Regarding the SMA-α-YOLOv5 network structure, Section 4.1 introduced the Stem block, and Section 4.2 presented the MNtV3-CA block, which is used to build the backbone network of YOLOv5. Additionally, the second to twelfth layer network structures in the MobileNetV3-Small [34] specification are used for reference. Finally, the loss function is optimized, and the improved structure is illustrated in Figure 7.

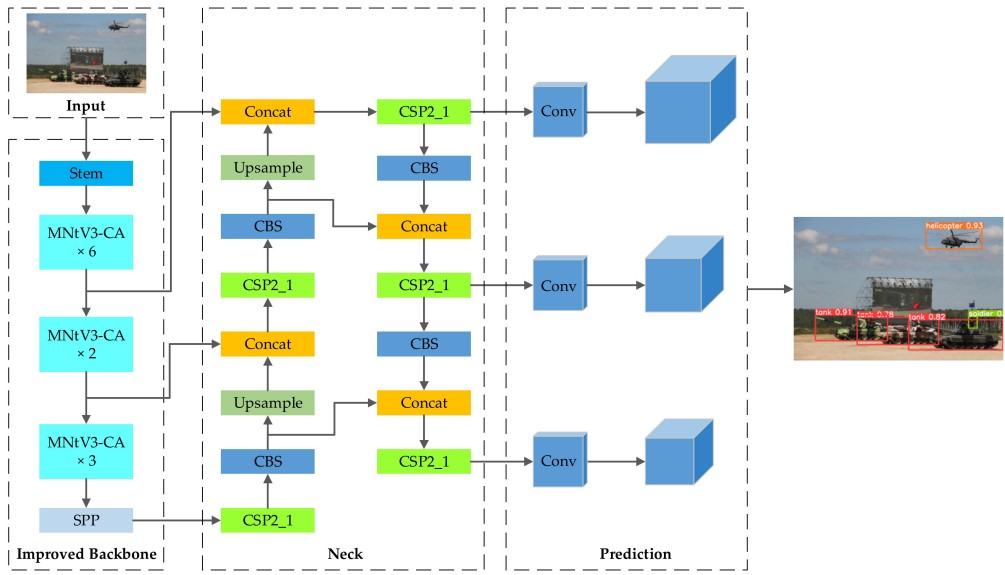

**Figure 7.** The network structure of SMCA-α-YOLOv5.

## 5. Experiments and Results

### 5.1. Experiment Platform

The experiments in this paper are carried out on the Google Colab development platform, and the experimental environment is Python3.6, Pytorch1.11.0, CUDA11.2, and Tesla V100-SXM2-16G. Data training, validation, and testing are performed with the same hyperparameters. Among them, the number of iterations is set to 100, the learning rate is set to 0.01, the initial learning rate momentum is 0.937, the weight decay coefficient is 0.0005, and the batch size is 64.

### 5.2. Evaluation Indicators

In order to verify the validity of the proposed model, a comprehensive evaluation is carried out using four indicators: mean average precision (mAP), model parameters (Parameters), model operation (GFLOPs), and detection speed (FPS). The average precision rate (AP) is the detection accuracy rate of a single target, composed of the area enclosed by the recall rate and the accuracy rate. The average precision is the average of all categories of AP values [24] and is used to evaluate the comprehensive detection performance of the model; the number of model parameters obtained during the model training process directly determines the size of the model file and the memory resources that the model consumes. The number of computations required throughout the model training process is referred to as the model computation volume, which directly represents the model's requirement for the hardware platform's computing capacity. The number of image data the model can detect per second is referred to as detection speed, and it is used to measure the model's performance in real time.

### 5.3. Analysis of Ablation Experiments

5.3.1. Ablation Experiment of Backbone Network

To verify the effectiveness of the developed algorithm, we conduct six groups of ablation experiments on the MITD dataset, while the YOLOv5s in Ultralytics 5.0 is used as the benchmark algorithm. The input image pixel size is set to $640 \times 640$, and the number of training iterations to 100.

The ablation results for each component are reported in Table 2, revealing that introducing the Stem block and MobileNetV3 block (MNtV3) increases inference time. However, the network is more lightweight in structure. The SENet attention mechanism in No. 3 is replaced with a CBAM and a CA attention mechanism in Nos. 4 and 5, respectively. Among them, the MobileNetV3 block utilizing the stem block and the embedded CA attention mechanism presents the best detection performance. Compared with YOLOv5, the mAP value of this model increased by 1.3%. The amount of parameters and computation amount decreased by 85.52% and 95.8%, respectively, and the average inference time increased by 0.003 sec. Although the detection speed is slightly reduced, other performances have been greatly improved, so the overall effect is the best.

**Table 2.** Results of ablation experiments.

| No | Model | mAP@0.5 | Parameters/$10^6$ | GFLOPs /$10^9$ | Inference Time /$s$ | FPS |
|----|-------|---------|-------------------|----------------|---------------------|-----|
| 0 | YOLOv5s | 96.5 | 7.070 | 16.4 | **0.019** | **52.6** |
| 1 | YOLOv5s+Stem | 95.9 | 4.502 | 5.8 | 0.022 | 45.5 |
| 2 | YOLOv5s+MNtV3 | 96.8 | 3.558 | 6.3 | 0.025 | 40 |
| 3 | YOLOv5s+Stem+MNtV3 | 96.6 | 1.384 | 0.7 | 0.023 | 43.5 |
| 4 | YOLOv5s+Stem+MNtV3-CBAM | 97.3 | **1.016** | 0.71 | 0.024 | 41.7 |
| 5 | SMCA-YOLOv5 | **97.8** | 1.024 | **0.69** | 0.022 | 45.5 |

The best results of every metric are bolded.

Figure 8 shows the improved PR curve of the YOLOv5s model. It can be seen that the improved model has achieved better detection results for various military targets.

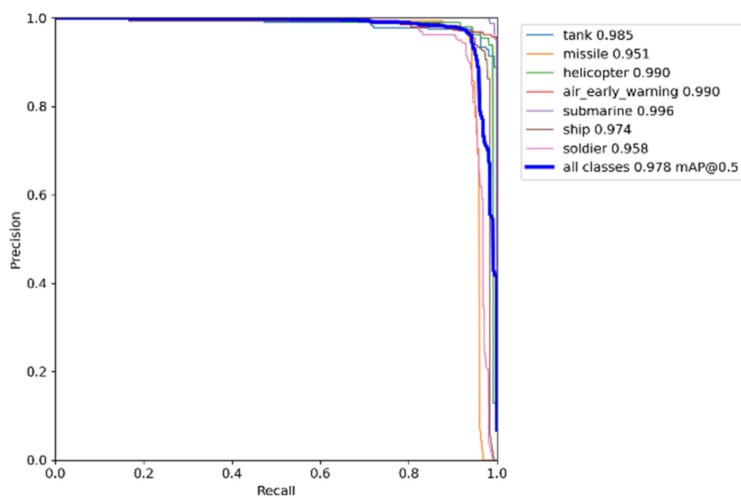

**Figure 8.** PR curve graph of the improved YOLOv5s model.

5.3.2. Ablation Experiment of Loss Function

To demonstrate the effectiveness of the proposed loss function, we perform five sets of ablation experiments on YOLOv5s and SMCA-YOLOv5, with the corresponding experimental results reported in Table 3. For fairness, after extensive experiments, we set the parameters to $\alpha = 3$ and $\gamma = 0.5$, which afford the best performance.

**Table 3.** Ablation experiment results under different losses.

| Loss | YOLOv5s Algorithm (mAP/%) | SMCA-YOLOv5 Algorithm (mAP/%) |
|---|---|---|
| $L_{CIOU}$ | 96.5 | 97.8 |
| $L_{EIOU}$ | 96.6 | 98.0 |
| $L_{\alpha\text{-EIOU}}$ | 96.8 | 97.9 |
| $L_{Focal\text{-}EIOU}$ | 97.0 | 98.2 |
| $L_{Focal\text{-}\alpha\text{-}EIOU}$ | 97.3 | 98.4 |

Figure 9 illustrates the effect of five loss functions on the SMCA-YOLOv5 algorithm. The Focal-$\alpha$-EIOU Loss has a better convergence speed and regression accuracy than the other four losses. The dataset in this paper has more high-quality samples. Thus, Focal-$\alpha$-EIOU Loss uses the weight of IOU$^{\alpha}$ to focus on high-quality samples. Therefore, when there are more high-quality samples, the convergence speed is faster and the regression accuracy is lower.

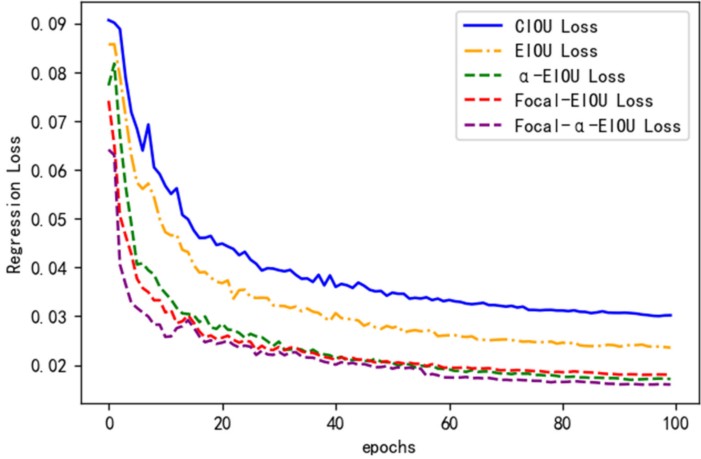

**Figure 9.** The effect of different losses on SMCA-YOLOv5.

To further verify the effectiveness of the proposed Focus-α-EIOU loss, the YOLOv5 algorithm is used to conduct experiments with different loss functions on the PASCAL VOC dataset and the CGMU dataset. We use the training and validation sets of the PASCAL VOC2007 and PASCAL VOC2012 datasets as training sets (containing 16,551 images in 20 categories) and the PASCAL VOC2007 test set (containing 4952 images) for testing. The results of the PASCAL VOC2007 dataset are reported in Table 4. Moreover, we use the training and validation sets of the CGMU dataset as a training set (containing 8007 images) and its test set (containing 1000 images) for testing. The results of the CGMU dataset are reported in Table 5. The experimental results in Tables 4 and 5 show that the proposed Focal-α-EIOU Loss attains the best performance on the AP55, AP60, AP95, and mAP metrics. Overall, the proposed loss outperforms the competitor's horizontal box regression losses.

**Table 4.** PASCAL VOC2007 test detection results. mAP denotes $mAP_{50:95}$.

| Loss Method | $AP_{50}$ | $AP_{55}$ | $AP_{60}$ | $AP_{65}$ | $AP_{70}$ | $AP_{75}$ | $AP_{80}$ | $AP_{85}$ | $AP_{90}$ | $AP_{95}$ | mAP |
|---|---|---|---|---|---|---|---|---|---|---|---|
| $L_{IOU}$ * | 78.6 | 76.1 | 73.2 | 69.2 | 64 | 57.5 | 48.5 | 36.4 | 20.1 | 2.8 | 52.64 |
| $L_{GIOU}$ * | 78.5 | 76.3 | 73.2 | 68.6 | 63.7 | 57 | 48.9 | 36.4 | 21.1 | 3.4 | 52.71 |
| $L_{DIOU}$ * | 78.6 | 76.3 | 73.2 | 69.1 | 63.6 | 57.3 | 49.5 | 37.4 | 21.6 | 3.3 | 52.99 |
| $L_{IIOU}$ * | 78.7 | 76.5 | 73.5 | 69.3 | 64.1 | 57.8 | 50.2 | 37.5 | 21.4 | 3.2 | 53.22 |
| $L_{CIOU}$ | **78.9** | 76.3 | 73.4 | 69.1 | 63.8 | 57.6 | 50.3 | 37.4 | 21.2 | 3.1 | 53.11 |
| $L_{EIOU}$ | 78.7 | 76.5 | 73.3 | 68.9 | 63.9 | 57.7 | 50.5 | 36.7 | 21.4 | 3.2 | 53.08 |
| $L_{\alpha\text{-}EIOU}$ | 78.5 | 76.4 | 73.3 | 69.7 | 64.3 | 58.3 | 49.9 | 37.7 | **21.9** | 3.1 | 53.31 |
| $L_{Focal\text{-}EIOU}$ | 78.6 | 76.8 | 73.5 | **69.4** | 64.3 | 58.2 | **50.8** | 37.5 | 21.6 | 3.3 | 53.40 |
| $L_{Focal\text{-}\alpha\text{-}EIOU}$ | 78.8 | **76.9** | **73.6** | 69.6 | **64.6** | **58.6** | 50.7 | **38.2** | 21.8 | **3.6** | **53.64** |

* Indicates cited reference [22]. The best results of every metric are bolded.

**Table 5.** CGMU test detection results.

| Loss Method | $AP_{50}$ | $AP_{55}$ | $AP_{60}$ | $AP_{65}$ | $AP_{70}$ | $AP_{75}$ | $AP_{80}$ | $AP_{85}$ | $AP_{90}$ | $AP_{95}$ | mAP |
|---|---|---|---|---|---|---|---|---|---|---|---|
| $L_{IOU}$ * | 48.8 | 45.8 | 42.4 | 38.3 | 32.3 | 27.7 | 20.7 | 13.6 | 6.2 | 0.8 | 27.66 |
| $L_{GIOU}$ * | 49.6 | 46.0 | 42.9 | 38.6 | 34.2 | 28.5 | 22.1 | **14.7** | 5.6 | 1.0 | 28.32 |
| $L_{DIOU}$ * | 49.5 | 46.5 | 43.1 | 39.0 | 33.2 | 28.3 | 22.2 | 13.2 | 6.4 | 1.2 | 28.26 |
| $L_{IIOU}$ * | **52.1** | 48.6 | 44.6 | 40.5 | 35.5 | 28.0 | 18.5 | 11.6 | 4.6 | 0.6 | 28.46 |
| $L_{CIOU}$ | 50.9 | 47.2 | 43.4 | 40.3 | 34.6 | 27.9 | 21.4 | 13.4 | 6.0 | 1.3 | 28.64 |
| $L_{EIOU}$ | 49.9 | 48.2 | 43.7 | 39.6 | 35.2 | **29.5** | 21.8 | 14.1 | 5.6 | 1.2 | 28.88 |
| $L_{\alpha\text{-}EIOU}$ | 50.7 | 48.5 | 43.4 | 40.1 | **36.3** | 28.1 | 22.2 | 14.4 | 5.9 | 1.0 | 29.06 |
| $L_{Focal\text{-}EIOU}$ | 51.4 | 49.1 | 43.6 | 40.7 | 35.4 | 27.1 | 21.5 | 13.6 | **6.9** | 1.1 | 29.04 |
| $L_{Focal\text{-}\alpha\text{-}EIOU}$ | 51.3 | **49.5** | **44.8** | **40.8** | 36.2 | 28.2 | **22.7** | 14.1 | 6.1 | **1.3** | **29.50** |

* Indicates cited reference [22]. The best results of every metric are bolded.

### 5.4. Compare with Other Algorithms

The detection effect of the algorithm in this paper on military targets is further analyzed, as shown in Table 6. Compared with SSD, Faster R-CNN, YOLOv3 algorithm of Ultralytics 9.5.0 version, Pytorch_YOLOv4 of WongKinYiu, and YOLOv5 of Ultralytics 5.0 version, the average inference time of SSD algorithm is the fastest, and the other optimal indicators are proposed in this paper.

According to Table 6, the proposed SMCA-α-YOLOv5 has the highest mAP value, and the average detection speed is 19.1 Frames Per Second (FPS), 5.0 FPS lower than the SSD and the YOLOv5 algorithms. Additionally, the proposed model has significant advantages considering network parameters and computation complexity. Overall, the improved model of this paper not only improves the detection accuracy but also effectively realizes the lightweight of the network structure and meets the needs of military target detection with limited platform resources.

**Table 6.** Performance comparison results of different algorithms.

| Model | Image size | mAP@0.5 | Parameters /$10^6$ | GFLOPs /$10^9$ | Inference Time /s | FPS |
|---|---|---|---|---|---|---|
| SSD | 640*640 | 90.1 | 26.79 | 31.4 | **0.015** | **66.7** |
| Faster R-CNN | 640*640 | 88.5 | 60.61 | 284.1 | 0.173 | 5.7 |
| YOLOv3 | 640*640 | 96.3 | 61.53 | 154.8 | 0.035 | 28.6 |
| YOLOv4 | 640*640 | 96.8 | 52.53 | 128.5 | 0.040 | 25 |
| YOLOv5 | 640*640 | 96.5 | 7.070 | 16.4 | 0.019 | 52.6 |
| SMCA-$\alpha$-YOLOv5 | 640*640 | **98.4** | **1.014** | **0.67** | 0.021 | 47.6 |

The best results of every metric are bolded.

### 5.5. Analysis of Detection Results

In order to more intuitively reflect the performance of the proposed algorithm, representative images are selected from the MITD test set as the test objects, and the military target detection results of the SMCA-$\alpha$-YOLOv5 and YOLOv5 algorithms in different scenarios are analyzed. Figure 10 illustrates the detection result of scene 1, while Figure 10a presents one helicopter target and nine soldier targets. Figure 10b presents the detection result of the YOLOv5 algorithm, where two soldier targets are missed (marked by the yellow ellipse in Figure 10b). Figure 10c is the detection result of the SMCA-$\alpha$-YOLOv5 algorithm, where a soldier target is missed (marked by a yellow ellipse in Figure 10c). Figure 11 is the detection result of scene 2, and Figure 11b presents the detection result of the YOLOv5 algorithm, where a tank target and a soldier target are missed (marked by the yellow ellipse in Figure 11b). Figure 11c depicts the detection result of the SMCA-$\alpha$-YOLOv5 algorithm, where a soldier target is missed (marked by a yellow ellipse in Figure 11c). Although the improved algorithm has missed detection, it still has advantages compared to the YOLOv5 algorithm.

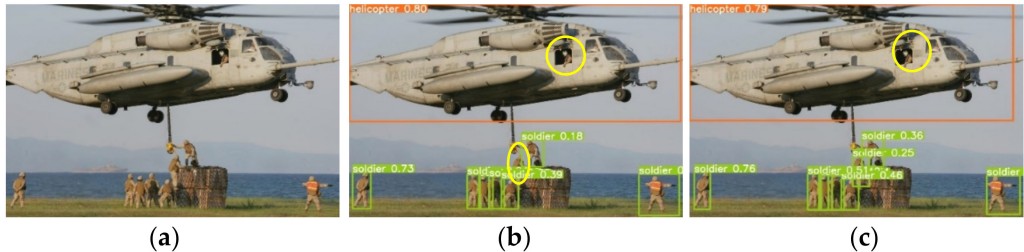

**Figure 10.** The detection result of scene 1. (**a**) Original image sample; (**b**) Detection results of the YOLOv5 algorithm; (**c**) Detection results of SMCA-$\alpha$-YOLOv5 algorithm.

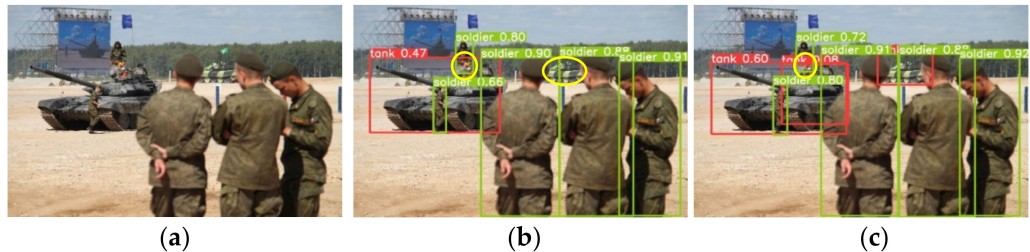

**Figure 11.** The detection result of scene 2. (**a**) Original image sample; (**b**) Detection results of the YOLOv5 algorithm; (**c**) Detection results of SMCA-$\alpha$-YOLOv5 algorithm.

In conclusion, introducing the Stem block and MobileNetV3 block into the backbone network reduces the network's parameters and computation complexity and increases the network structure's depth, thereby increasing the area of the receptive field. At the same time, combined with the lightweight coordinate attention mechanism, the network's feature extraction ability for occluded and small targets is enhanced further. Improving the



loss function affords the regression process to focus on high-quality anchor boxes, and thus the improved algorithm has strong robustness.

## 6. Conclusions

Aiming at the difficulty of deploying military target detection algorithms on embedded platforms with limited resources, a lightweight military target detection method based on improved YOLOv5 is proposed. This method redesigns the backbone network of YOLOv5 by introducing Stem block and MobileNetV3 block to reduce the number of parameters and computation of the model. In order to further improve the feature expression ability of the network, a coordinate attention module is embedded in the MobileNetV3 block structure, which improves the detection performance of the model for military targets. Based on EIOU Loss and Focal Loss, a loss with power parameter $\alpha$ is designed to optimize CIOU Loss, which provides more flexibility for the detector and achieves different levels of bounding box regression accuracy. The experimental results show that the algorithm proposed in this paper can ensure real-time performance and detection accuracy, and can also meet the needs of military target detection under the condition of limited resources of weapon equipment platforms.

The experimental results show that the average inference time of the algorithm proposed in this paper has increased, and the next step is to use the pruning algorithm to compress the backbone network composed of Stem block and MNtV3-CA block to improve the average detection speed. At the same time, the algorithm is deployed on embedded devices with limited hardware resources to verify the applicability of the algorithm in this paper.

**Author Contributions:** Conceptualization, L.S., X.D., S.Q. and Y.L.; methodology, L.S.; software, L.S. and S.Q.; validation, L.S., X.D. and Y.L.; formal analysis, L.S. and S.Q.; investigation, L.S. and S.Q.; resources, X.D. and Y.L.; data curation, L.S. and S.Q.; writing—original draft preparation, L.S.; writing—review and editing, X.D. and Y.L.; visualization, L.S.; supervision, X.D. and Y.L.; project administration, X.D. and Y.L.; funding acquisition, X.D. and Y.L. All authors have read and agreed to the published version of the manuscript.

**Funding:** This work was financially supported by the "Liaoning BaiQianWan Talents Program" (Grant No. 2018921080).

**Data Availability Statement:** The data presented in this study are available on request from the corresponding author.

**Conflicts of Interest:** The authors declare no conflict of interest.

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
