# Peer review of "A Lightweight Military Target Detection Algorithm Based on Improved YOLOv5"

_electronics, doi:10.3390/electronics11203263_

Round 1

Reviewer 1 Report

1. More background  review  to be included.

2. Detailed explanation to be required  about the results. Example consider 10(b),10(c) etc.

3.English language presentation to be improved.

4. Conclusions: Include/suggest future  inference time, FPS improvement methodology in  your proposed method.

Reviewer 2 Report

The authors have prepared a new dataset to detect military objects in this research. The YOLOv5 architecture has been modified with the MobileNet backbone, stem block, and CA module. A new loss function has been proposed to improve the model's accuracy. I have the following reviews for the article. 

1.  The authors have proposed the usage of Focal-EIOU with power Alpha to enhance performance. But they have not explained why? What are the drawbacks of Focal-EIOU and Alpha-IoU? What benefits does the proposed metric provide? A detailed explanation is needed in section 4.3.  

A literature review can be added for various loss functions. 

Table 4 provides increased mAP, but the improvement is not significant. When a new loss metric is proposed, it should be experimented on various datasets, including the benchmark COCO and PASCAL datasets. This is necessary to visualize the performance in dense objects and how the loss function performs when many object categories are present. Additional experiments need to be conducted. 

Below are the latest loss functions to solve the bounding box regression. They can be included in the literature review. 

a. Piou loss: Towards accurate oriented object detection in complex environments

b. Zhou D, Fang J, Song X, et al. Iou loss for 2d/3d object detection[C]//2019 International Conference on 3D Vision (3DV). IEEE, 2019: 85-94

c. Ravi, N.; El-Sharkawy, M. Real-Time Embedded Implementation of Improved Object Detector for Resource-Constrained Devices. J. Low Power Electron. Appl. 2022, 12, 21. https://doi.org/10.3390/jlpea12020021

d. Yang X, Yan J, Ming Q, et al. Rethinking rotated object detection with gaussian wasserstein distance loss[C]//International Conference on Machine Learning. PMLR, 2021: 11830-11841

2. Stem and CA modules have been implemented in various research articles. This research work utilizes the module and mobilenet to reduce the model size and increase mAP. There is not innovative idea. Please explain why the contributions are significant and how they differ from other research works in yolov5?

3. Various literature have been published on YOLOv5, which improves accuracy and enhances performance. How is this research significant/different from others? 

4. Section 5 content can be reduced. The tables and images are self-explanatory to understand the superiority of the proposed research. Only significant details can be explained.  

Round 2

Reviewer 2 Report

The authors have addressed all my concerns.